# Pharmacological Cardioprotection against Ischemia Reperfusion Injury—The Search for a Clinical Effective Therapy

**DOI:** 10.3390/cells12101432

**Published:** 2023-05-20

**Authors:** Qian Wang, Coert J. Zuurbier, Ragnar Huhn, Carolin Torregroza, Markus W. Hollmann, Benedikt Preckel, Charissa E. van den Brom, Nina C. Weber

**Affiliations:** 1Department of Anesthesiology—L.E.I.C.A., Amsterdam University Medical Centers, Location AMC, Cardiovascular Science, Meibergdreef 11, 1105 AZ Amsterdam, The Netherlands; q.w.wang@amsterdamumc.nl (Q.W.); c.j.zuurbier@amsterdamumc.nl (C.J.Z.); b.preckel@amsterdamumc.nl (B.P.); c.vandenbrom@amsterdamumc.nl (C.E.v.d.B.); n.c.hauck@amsterdamumc.nl (N.C.W.); 2Department of Anesthesiology, Kerckhoff-Clinic-Center for Heart, Lung, Vascular and Rheumatic Disease, Justus-Liebig-University Giessen, Benekestr. 2-8, 61231 Bad Nauheim, Germany; r.huhn-wientgen@kerckhoff-klinik.de (R.H.); c.torregroza@kerckhoff-klinik.de (C.T.)

**Keywords:** cardioprotection, ischemia-reperfusion injury, drug development

## Abstract

Pharmacological conditioning aims to protect the heart from myocardial ischemia-reperfusion injury (IRI). Despite extensive research in this area, today, a significant gap remains between experimental findings and clinical practice. This review provides an update on recent developments in pharmacological conditioning in the experimental setting and summarizes the clinical evidence of these cardioprotective strategies in the perioperative setting. We start describing the crucial cellular processes during ischemia and reperfusion that drive acute IRI through changes in critical compounds (∆G_ATP_, Na^+^, Ca^2+^, pH, glycogen, succinate, glucose-6-phosphate, mitoHKII, acylcarnitines, BH_4_, and NAD^+^). These compounds all precipitate common end-effector mechanisms of IRI, such as reactive oxygen species (ROS) generation, Ca^2+^ overload, and mitochondrial permeability transition pore opening (mPTP). We further discuss novel promising interventions targeting these processes, with emphasis on cardiomyocytes and the endothelium. The limited translatability from basic research to clinical practice is likely due to the lack of comorbidities, comedications, and peri-operative treatments in preclinical animal models, employing only monotherapy/monointervention, and the use of no-flow (always in preclinical models) versus low-flow ischemia (often in humans). Future research should focus on improved matching between preclinical models and clinical reality, and on aligning multitarget therapy with optimized dosing and timing towards the human condition.

## 1. Introduction

The present article describes the pivotal cellular processes occurring during ischemia and reperfusion that contribute to acute IRI and provides an up-to-date overview of novel and promising interventions that target these processes. We focus on cardiomyocytes and the endothelium. The review highlights the need to address the limited translatability of findings from basic research to clinical practice, which may be attributed to factors such as the absence of comorbidities, concurrent medications, and peri-operative treatments in preclinical animal models. In order to bridge this gap, future research should prioritize improved alignment between preclinical models and clinical reality, considering factors such as comorbidities, comedications, and peri-operative treatments.

The review focuses purposely only on novel pharmacological interventions, thus not on any “ischemic conditioning” protocol or drugs that have been tried already in several clinical trials, such as, e.g., cyclosporine A, nitrite, antioxidants, or gene therapies. It is also not the intention of this review to cover all IR mechanisms and all protective strategies reported in the literature. Such an extensive review was already recently published [1]. Additionally, we mainly focus on novel compounds for which the on-target IR mechanisms are clearly defined, such as NAD^+^ precursors to restore the IR-induced depletion of NAD+, or inhibitors of succinate oxidation (acidic malonate) to reduce the succinate-driven ROS production at early reperfusion. We have left out other novel promising compounds that have a much broader but more undefined mechanism(s) of protection, such as microRNAs [2,3], SGLT2i’s [4,5], and extracellular vesicles [6,7].

## 2. Mechanisms of Cardiac IRI

Myocardial IRI is, first of all, caused by the alteration of several specific cellular processes during the actual ischemic episode. Early restoration of flow to ischemic tissue, i.e., reperfusion, which is clinically achieved by thrombolytics, percutaneous coronary intervention, or other revascularization modalities, prevents further ischemic cardiomyocyte death, but paradoxically results in further damage of the cardiac tissue [8]. Important cellular processes during both the ischemic and reperfusion period that are known to contribute to acute cardiac IRI will be discussed and summarized in Figure 1. We will mainly focus on the first hours of cardiac IRI because, in this period, most of the damage occurs.

During ischemia, the lack of oxygen causes ΔG_ATP_ to decrease, resulting in increased acidosis and the build-up of intracellular Na^+^ and, consequently, Ca^2+^. In the mitochondria, succinate accumulates from the breakdown of TCA intermediates and glycogen. The decreased pH, together with the build-up of G6P from glycogen breakdown, detach HKII from mitochondria, facilitating mitochondrial ROS production. The Ca^2+^-activated NOX, together with eNOS uncoupling driven by BH4 oxidation, increase cytosolic ROS production. Mitochondrial acylcarnitines accumulate due to the activation of CPT1. At reperfusion, the efflux of H^+^ through reactivation of NHE results in a burst of cytosolic Na^+^ and Ca^2+^. Accumulated succinate drives mitochondrial ROS production through RET. HKII detachment from mitochondria, high mitochondrial acylcarnitine inhibition of the ETC, and Ca^2+^-induced NOX contribute to ROS production. Following pH normalization, the high ROS and Ca^2+^ induce mPTP opening, allowing NAD^+^ and CytC to escape from mitochondria and dissipating mitochondrial membrane potential. NAD^+^ is degraded by NADases into the Ca^2+^-mobilizing second messenger cADPR, and cytC activates apoptosis. Additionally, high Ca^2+^ causes hypercontracture, calpain, and caspase activation. Acytel-CoA = acetyl coenzyme A; ADP = adenosine diphosphate; ATP = adenosine triphosphate; BH_4_ = tetrahydrobiopterin; cADPR = cyclic ADP-ribose; Casp = caspase; Cyt C = cytochrome C; CTP1 = carnitine palmitoyltransferase 1; eNOS = endothelial nitric oxide synthase; FAD = flavin adenine dinucleotide; FADH2 = reduced flavin adenine dinucleotide; Gly = glycogen; G6P = glucose 6-phosphate; HK2 = hexokinase 2; mPTP = mitochondrial permeability transition pore; NAD^+^ = oxidized nicotinamide adenine dinucleotide; NADH = reduced nicotinamide adenine dinucleotide; NCX = Na^+^/Ca^2+^ exchanger; NHE = Na^+^/H^+^ exchanger; NKA = Na^+^/K^+^-ATPase; NLRP3 = nucleotide-binding domain, leucine-rich–containing family, pyrin domain–containing-3; NOX = nicotinamide adenine dinucleotide phosphate oxidase; RET = reverse electron transfer; ROS = reactive oxygen species; SERCA = sarco-endoplasmic reticulum calcium ATPase; SR = sarcoplasmic reticulum; 4-HNE = 4-hydroxynonenal; 3-NT = 3-nitrotyrosine; and α kg = alpha ketoglutarate.

### 2.1. Decreased ΔG_ATP_, Acidosis, and Ion Disturbances as Initial Processes Driving IRI

Myocardial IRI starts from the onset of ischemia. During ischemia, cardiomyocyte metabolism shifts from oxidative phosphorylation to anaerobic glycolysis because of the lack of oxygen and nutrients. The increased glycolytic ATP turnover, together with the diminishing mitochondrial ATP synthesis, CO_2_ accumulation, and net ATP breakdown, increases the build-up of H^+,^ resulting in intracellular acidosis [9]. Ion homeostasis is largely dictated by thermodynamic control of the active ion pumps (e.g., the sodium/potassium ATPase (NKA pump)) and exchangers (e.g., the sarcoplasmic–endoplasmic reticulum calcium-ATPase (SERCA)), i.e., determined by the amount of free energy that can be liberated during ATP hydrolysis (ΔG_ATP_). During the first ten minutes of ischemia, a first initial small drop in ΔG_ATP_ that starts to inhibit the pumps, and that causes intracellular Na^+^ to increase and intracellular K^+^ to decrease, is observed [10]. Then, following the depletion of glycogen and halting of anaerobic glycolysis, ΔG_ATP_ starts to fall much more, driving further increases in Na^+^ and decreases in K^+^ [10]. The increase in Na^+^ drives the rise in intracellular Ca^2+^ by impairing the normal forward mode action (exchanging intracellular Ca^2+^ for extracellular Na^+^) of the sodium–calcium exchanger (NCX). Ischemic contracture starts to develop because of rising cytosolic calcium and lack of ATP [11]. Intracellular acidosis may further increase intracellular Na^+^ load through activation of the sodium–hydrogen exchanger (NHE), although the quick development of extracellular acidosis during ischemia will start to inhibit NHE activity [12,13]. At reperfusion, extracellular acidosis is quickly normalized, reactivating NHE to supranormal activity [12] to restore intracellular pH at the expense of further increasing intracellular Na^+^. The rise in intracellular Na^+^ further impairs or even reverses the NCX, extruding Na^+^, now at the expense of a large Ca^2+^ overload. 

This burst in cytosolic Ca^2+^ within the first minutes of reperfusion, together with the fast normalization of intracellular pH, is one of the crucial mediators of IRI. The high calcium level facilitates (1) opening of the mitochondrial permeability transition pore (mPTP), dissipating mitochondrial potential and thereby ATP synthesis, causing swelling and rupture of mitochondria and consequently cardiac cells; (2) activation of calpains, proteolytic proteases that breakdown the cell, and activates caspases through dislodgement of these programmed cell death enzymes from the cytoskeleton, resulting in pyroptosis through an NLRP3 dependent or independent mechanism [14,15]; and (3) hypercontracture of the myofilaments and thereby mechanical rupture of cardiac tissue, also resulting in cell death [9,16].

### 2.2. Oxidative Stress Involvement in IRI

Part of IR-associated oxidative stress develops during the ischemic period, as demonstrated by techniques allowing for measurements of direct ROS production [17,18,19,20]. These findings are supported by end-ischemic measurements of cardiac 4-hydroxy-2-nonenal (4-HNE), a breakdown product of lipid peroxidation, and 3-nitrotyrosine, a marker of nitrated tyrosine due to peroxinitrite, both specific biomarkers of oxidative stress [21]. Ischemic ROS mostly develops during the initial period of ischemia, but not during later ischemia [21,22,23]. Interestingly, once 4-HNE is generated during ischemia, no further 4-HNE is produced during reperfusion, and reperfusion in itself cannot reduce the ischemic build-up of 4-HNE. Sources of ROS during ischemia include increasing cytochrome p450 activity, mitochondrial electron transport damage and uncoupling, NAD(P)H oxidase (NOX) activity, and uncoupled nitric oxide synthase (NOS) activity [24]. In addition, oxidative damage during ischemia can also be a result of a decreased antioxidant capacity in the ischemic heart, creating a more pro-oxidant environment. Reduced glutathione (GSH) is an important endogenous antioxidant in the heart that is rapidly oxidized during ischemia. Restoration into GSH needs ATP, which becomes scarce.

Next to ischemic ROS, a burst of ROS is also initiated during the first minutes of reperfusion [17,25,26]. A major contributor to the ROS burst during reperfusion is the oxidation of the ischemic accumulated succinate (see Section 2.3.1 below) [27]. Although two-thirds of the ischemic succinate is released from the heart upon reperfusion, the remaining one-third is oxidized during the first five minutes of reperfusion. At the end of ischemia, the heart is already in a highly reduced state, with maximal high levels of NADH and a maximally reduced ubiquinone (Q) pool. At this early stage of reperfusion, succinate is rapidly oxidized to fumarate by succinate dehydrogenase (SDH), trying to further reduce the already highly reduced Q pool. Because the Q pool is already maximally reduced, electrons are forced backward (reverse electron transport: RET), and O_2_ is partially reduced to O^2−^ at complex I [27]. Other contributors to reperfusion ROS are activated NOX2/4 [28], highly charged (high ψm) mitochondria due to detachment of Hexokinase II (HKII) from mitochondria [26,29,30,31], interrupted oxidative phosphorylation due to ischemic accumulation of long-chain acylcarnitines (see below Section 2.3.3) [32,33], and uncoupled endothelial NOS (eNOS) [24,34].

ROS produced during early reperfusion primes the mPTP for the high cytosolic calcium, permanently opening this mitochondrial pore and causing cell death [35]. ROS and nitrative stress, both during ischemia and reperfusion, also oxidizes and/or nitrates several proteins, thereby hampering their function and contributing to IRI. 

### 2.3. Involvement of Intermediate Metabolites in IRI

IRI of the heart can be considered a metabolic pathology with abruptly halted metabolism during ischemia, and is aggravated by a sudden restart of specific metabolic pathways trying to recover metabolic homeostasis at reperfusion [36]. Variations in cellular levels of metabolic intermediates during IR underlie part of the mechanisms contributing to the development of IRI. Here, we describe changes in metabolic intermediates that are now known to affect cardiac IRI. 

#### 2.3.1. Succinate

It has been known since 1978 [37] that succinate increases in the heart during ischemia. More recent research demonstrated that oxidation of ischemic succinate during the first minutes of reperfusion contributes largely to early reperfusion ROS and, therefore, IRI [27]. Originally it was suggested, by employing isolated mitochondria as a model, that ischemic succinate was generated from fumarate through mitochondrial complex II reversal [27]. However, research using a more physiological model of isolated mouse hearts demonstrated that ischemic succinate is mostly arising from remaining Krebs cycle activity during ischemia, still turning in the normal direction (so-called canonical TCA activity) [38]. The metabolic precursors of ischemic succinate are mostly (75%) intermediates of the Krebs cycle, whereas the remaining 25% is generated through complex II reversal (6%), aspartate anaplerosis (10%) via aspartate aminotransferase (AST), and glutamine anaplerosis (8%) via alanine aminotransferase (ALT) [38]. Increased glycogen breakdown and glycolysis during ischemia indirectly contribute to ischemic succinate accumulation by providing the substrates (pyruvate, NADH) for AST and ALT. Actually, the depletion of glycogen before ischemia resulted in a 40% decrease in ischemic succinate, demonstrating the importance of glycogen during ischemia for the generation of ischemic succinate [38]. Of note, the generation of succinate during ischemia can also be protective against IRI, as it provides energy in the form of GTP through substrate-phosphorylation by succinyl-coenzyme A (CoA) synthetase. Those beneficial effects were reflected by delayed contracture with increased succinate during ischemia [38]. Thus, impairing succinate generation during ischemia is a double-edged sword, whereas impairing its oxidation during early reperfusion is mostly protective against IRI.

#### 2.3.2. Glycogen and Glucose-6-Phosphate

Several glycolytic intermediates and end-products accumulate during ischemia from the breakdown of glycogen, such as glucose-1-phosphate, glucose-6-phosphate (G6P), fructose-6-phosphate, pyruvate, and lactate [39,40]. Although for several of these intermediates, it remains to be elucidated whether they modulate IRI; elevated G6P levels are detrimental. The increase in G6P, together with acidosis during ischemia, will detach HKII from the mitochondria, thereby increasing infarct size [41,42]. The amount of HKII bound to mitochondria (mtHKII) is a strong determinant of infarct size [41], making treatments directed at maintaining mtHKII a valid target for cardioprotection [43]. The strongest cardioprotective intervention to date, ischemic preconditioning (IPC), was shown to be associated with differences in mtHKII, but not with, e.g., ischemic succinate accumulation [41,42,43,44,45]. Maintaining HKII at the mitochondria may provide protection through several mechanisms: (1) reduction in the mitochondrial membrane potential to lower mitochondrial ROS production [29], (2) activation of glucose metabolism with attenuation of mitochondrial activity [46], (3) impairment of mPTP opening [47], and (4) prevention of ischemic mitochondrial ATP hydrolysis by reversed ATPase [48], thereby also attenuating anaerobic glycolysis and acidosis during ischemia. Pretreating hearts with insulin may therefore increase cardiac IRI, because insulin will load the heart with glycogen, which, once totally depleted because of prolonged ischemia, increases G6P and acidosis, detaching HKII from mitochondria and increasing infarct size [49]. However, when total glycogen depletion is prevented, for example, due to such short ischemia that glycogen is not totally depleted during ischemia, insulin pretreatment is protective because insulin per se increases cardiac mtHKII [50] and the increased glycogen provides additional glycolytic ATP during ischemia [51]. Therefore, glycogen is also considered a double-edged sword during IRI: protective when not completely depleted, detrimental when all glycogen is broken down during ischemia [51]. 

#### 2.3.3. Acylcarnitines

Older literature has reported the accumulation of acyl-CoA and acylcarnitine during ischemia [52,53]. Recent studies demonstrated especially the accumulation of long-chain acylcarnitines to contribute to cardiac IRI [32,36]. Acylcarnitines accumulate during ischemia because carnitine palmitoyltransferase 1 (CPT1, the enzyme catalyzing the transfer of acyl-coA to carnitine to form acylcarnitine) becomes highly activated due to the decrease in its natural inhibitor malonyl-CoA [54]. The enzyme-generating malonyl-CoA, acetyl-CoA carboxylase, is inhibited due to the ischemia-induced increase in phospho-AMPK. In turn, CPT1 generates large amounts of long-chain acylcarnitines that cannot be metabolized by mitochondria due to the lack of oxygen [32]. This detrimental effect of high cardiac levels of acylcarnitines on IRI is especially present in the fasted condition, when acylcarnitines can accumulate five times more than in the fed condition [55]. Mechanistically, the presence of high levels of long-chain acylcarnitines inhibits oxidative phosphorylation (OXPHOS) during early reperfusion, thereby inducing mitochondrial membrane hyperpolarization and contributing to mitochondrial ROS production [56] to increase cardiac IRI. Thus, the detrimental effects of long-chain acylcarnitines become apparent during reperfusion, when the supply of oxygen drives ROS from acylcarnitine-impaired mitochondria [33].

### 2.4. Cofactors Involved in IRI

Tetrahydrobiopterin (BH4): For nitric oxide (NO) synthesis, eNOS requires the redox-sensitive cofactor BH4. However, this cofactor is continuously oxidized during ischemia, such that at 30 min of ischemia, 85% of BH4 is already irreversibly degraded [34]. As a consequence, eNOS becomes uncoupled, and the enzyme starts to generate ROS (•O^2−^) instead of NO. It was estimated that ROS produced by uncoupled eNOS following 30 min of ischemia and reperfusion contributed to 30% of the total ROS detected [55]. Restoring BH4 directly by supplying BH4 in the liposomal formulation [34], or indirectly through, e.g., folic acid administration, protected hearts against IRI [57,58]. 

Nicotinamide adenine dinucleotide (NAD^+^): NAD^+^ is a critical cofactor involved in more than 500 enzymatic reactions impacting cellular metabolism, inflammation, energetics, and cell survival, with decreased levels related to a diverse area of pathologies such as aging, diabetes, metabolic diseases, heart failure, and IRI [59,60,61]. In the heart, 72% of all NAD^+^ is localized in the mitochondrial matrix [62]. During cardiac ischemia, about 30% of NAD^+^ is lost, which is largely increased to 70% during reperfusion [62]. The loss of NAD^+^ during reperfusion can be explained by the opening of the mitochondrial permeability transition pore (mPTP), a phenomenon mainly occurring during reperfusion and allowing NAD^+^ to escape the mitochondria. Outside the mitochondria, NAD^+^ is broken down by NADases (e.g., the glycohydrolase enzyme CD38) [62,63] to generate Ca^2+^-mobilizing second messenger cyclic ADPR (cADPR) to raise Ca^2+^ and thus contribute to IRI. Inhibiting the enzyme aldose reductase of the polyol pathway may be another means to conserve NAD^+^ during IR by attenuation of NAD^+^ use through reduced sorbitol dehydrogenase activity. The preserved NAD^+^ is then used for glyceraldehyde 3-phosphate dehydrogenase in glycolysis, facilitating increases in glycolytic ATP synthesis [64,65,66]. Major mechanisms explaining increased cardiac IRI with decreased NAD^+^ relates to the loss of function of enzymes that need NAD^+^ as a cofactor, such as sirtuins (deacetylation) and glycolytic enzymes. Decreased sirtuin and glycolytic activities are known to increase cardiac IRI [36,67,68]. 

## 3. Novel Pharmacological Strategies Targeting IRI in Cardiomyocytes

Drugs modulate their cardioprotection effects through different upstream routes; however, their final end-effector protective mechanisms culminate in the attenuation of Na^+^ and Ca^2+^ overload, inhibition of ROS production, and mPTP opening, precipitating reductions in cell death [69]. We now focus on novel pharmacological treatments that mainly target the mechanisms discussed above in Section 2.

### 3.1. NAD^+^ Precursors

NAD^+^ precursors that have been tested to prevent cardiac IRI ex vivo or in vivo are nicotinamide mononucleotide (NMN) and nicotinamide riboside (NR). One of the first studies examining NAD^+^ precursors in the setting of cardiac IRI [67] reported protection with 500 mg/kg NMN intraperitonally (i.p.), either once 30 min before ischemia, or four times just before and during 24 h of reperfusion. Mechanisms of protection by NMN precursors relate to the activation of sirtuins, glycolysis, and/or autophagy [67,68,70,71]. Surprisingly, few studies have examined NR as an NAD^+^ precursor in cardiac IRI studies, which is somewhat surprising given NR’s superiority in elevating NAD^+^ in humans [72]. Pre-ischemic administration (50 mg/kg i.v.) of NR in an in vivo rat model of cardiac IRI employing clinically relevant anesthesia reduced infarct size [73]. Subsequently, employing isolated mouse hearts, we also demonstrated that NR’s protection against cardiac IRI, similar to NMN [68], is mediated through glycolysis activation [74]. Thus, it seems that NAD^+^ precursors offer novel opportunities for acute protection against cardiac IRI. Although studies in isolated mitochondria have suggested protection through direct inhibition of mPTP by NAD^+^ [75], studies in intact hearts or in vivo favor activation of glycolysis as the major protective mechanism of these compounds.

### 3.2. Malonate

Malonate is a three-carbon dicarboxylic acid and a competitive succinate dehydrogenase (SDH) inhibitor, and as such, it inhibits succinate oxidation during early reperfusion [27]. Valls-Lacalle et al. found that disodium malonate reduced infarct size without increasing the incidence of ventricular fibrillation in pigs with left anterior descending coronary artery occlusion. No malonate was detected in distant myocardium or in plasma [76]. The authors attributed the observed cardioprotective effect of disodium malonate to intracoronary administration, which is a clinically feasible method to achieve selective delivery of cardioprotective treatments. However, other studies reported a lack of malonate protection, especially under diabetic conditions [77,78]. Although the reasons for this failure are not completely clear, it could have been partly explained by the absence of malonate in the hearts at the first minutes of reperfusion due to the experimental set-up. Recently, data showed that disodium malonate protects in an ischemia-selective way, i.e., only ischemic tissue with decreased pH and increased lactate will take up disodium malonate [79]. Accumulation of lactate and protons in ischemic tissue facilitates the protonation of malonate to its monocarboxylate form, which is exchanged for lactate through the monocarboxylate transporter 1 (MCT1) to quickly enter cardiomyocytes upon early reperfusion. Then, malonate is transported into mitochondria by the mitochondrial dicarboxylate carrier and subsequently inhibits succinate oxidation. Future research should focus on this type of malonate in preclinical models considering conditions and drugs that are present under clinical conditions (see Section 4). 

### 3.3. NLRP3 Inflammasome Inhibitors

The innate immune system is the first line of defense against exogenous (invading pathogens) or endogenous (sterile, such as traumatic impact, metabolic, or IR) stress signals, and is constituted of pattern-recognition receptors (PPRs) that are localized either on the plasma membrane and endogenous endosomes (Toll-like receptors; TLRs) or within the cytosol (Nucleotide-binding domain and Leucine-rich repeat-containing proteins; NLRs). Although TLR inhibition certainly holds promise for reducing cardiac IRI [80,81], the development of NLR inhibitors is currently more prominent [82]. 

The most prominent NLR suggested to be involved in acute cardiac IRI and for which inhibitors are being developed and tested is the NLRP3 inflammasome. NLRP3 inflammasome activity is regulated through expression and then oligomerization of three components: ASC, NLRP3, and caspase-1 [83]. NLRP3 is hardly expressed in healthy hearts, explaining the lack of cardioprotection following NLRP3 gene deletion [84,85]. However, stress and inflammation lead to NLRP3 expression in the heart. Stress induction can occur (1) during extensive surgery needed for opening the chest, (2) in diseased animals (diabetic, heart failure, metabolic syndrome, and aging), (3) in the presence of pathophysiology (low blood pressure, hypoxia, ischemia, and metabolic stress), or (4) examining hearts at late (>3 h) reperfusion. Impairing NLRP3 inflammasome activity through ASC deletion decreased infarct size after two days of reperfusion [86]. However, deletion of ASC or NLRP3 had no effect on IRI when studied between 30 min and 3 h of reperfusion [84,85]. Similarly, cardioprotection by NLRP3 inhibition was only observed at 24 h but not at 3 h of reperfusion [87]. Moreover, NLRP3 inhibition only reduced infarct size when administered 1 h after reperfusion, while its cardioprotective effect was lost when treatment was delayed to 3 h of reperfusion [87]. These findings highlight that the therapeutic window of NLRP3 inflammasome inhibition is limited to the first hours of reperfusion.

In a large animal model of cardiac I/R, the NLRP3 inhibitor MCC950 administered during the first 7 days of reperfusion exerted a mild protective effect on infarct size [88]. However, cardioprotection was not observed with the novel NLRP3-inflammasome inhibitor IZD334 [89]. No clinical trials have been published reporting on the use of specific NLRP3 inflammasome inhibitors in the setting of acute IRI. Only the old and aspecific anti-inflammatory drug colchicine, which inhibits the assembly of the NLRP3 inflammasome, was recently evaluated in a clinical trial. Colchicine was administered immediately at reperfusion and during the first five days of reperfusion in first-time ST-segment-elevation myocardial infarct patients [90]. However, no cardioprotection was observed. Although interleukin-6 and high-sensitivity C-reactive protein concentrations after PCI were decreased by colchicine in another study, the myocardial injury did not differ between the treatment and placebo groups [91]. 

In conclusion, although preclinical research shows promise for NLRP3 inhibitors to reduce cardiac IRI in settings of diseased hearts (diabetic, aged, and metabolic syndrome), and novel NLRP3 inflammasome inhibitors are being developed, the first cardioprotective trial with these selective NLRP3 inhibitors is still eagerly awaited [91]. 

### 3.4. Caspase and Calpain Inhibitors

Caspases, or cysteine-aspartic acid proteases, are cysteine endoproteases that attack and cleave a protein only after an aspartic acid residue. Caspase-3, -6, -7, -8, and -9 are mainly involved in apoptosis, whereas caspase-1, -4, -5, and -12 (in humans) and caspase-1, -11, and -12 (in mice) are mainly involved in inflammatory pathways [92]. 

Although apoptosis is important for heart development, a resistance to caspase-dependent apoptotic cell death has been detected in differentiated cardiomyocytes [93]. In addition, caspase-3/caspase-7 double-knockout mice showed no myocardial protection during the acute phase of reperfusion [94]. Additionally, in terminally differentiated mouse myocardium, caspase-3, -6, and -7 are silenced [95]. Directly comparing necrosis and apoptosis in an isolated rabbit heart employing different durations of ischemia, it was observed that necrosis and infarct size (12–23% of heart tissue) was 6–8 times larger than apoptosis (2–3% of cells) [96]. Controversial data have been reported for selective inhibitors of caspase-3 or -9, with one study reporting significant reductions in infarct size [97], whereas another study reported no effects on infarct size [98]. It is possible that the selective caspase inhibitors are not so selective and also partly inhibit, e.g., calpains (see below). Those results demonstrate that pharmacological treatment targeting apoptosis-related caspases does probably not play a major role in acute cardiac IRI.

In contrast, inhibition of caspase-1 and -4 by VRT-043198 (VRT) or emricasan did reduce acute cardiac IRI in both rats and mice [15,99,100]. The degree of protection obtained with the specific caspase-1 and -4 inhibitor VRT was similar to the protection observed with the pan-caspase inhibitor emricasan, again providing evidence that apoptosis does not contribute significantly to acute cardiac IRI [15]. Interestingly, specific calpain inhibition offered similar protection as that observed with VRT, and adding VRT to calpain inhibition did not further reduce infarct size, suggesting that the activation of calpain during early reperfusion is needed for caspase-1-induced infarct size reduction [15]. Calpains are activated by cytosolic increases in Ca^2+^, one of the two major signaling events (besides ROS production) during the first minutes of reperfusion that dictate acute cardiac IRI (see sections above). Indeed, calpain silencing ameliorated acute cardiac IRI [101]. Although caspase-1 activation has originally been thought to be a consequence of NLRP3 inflammasome activation, recent research showed that calpains can activate caspase-1 through dislodging procaspase-1 from the actin-filament network and transformation of its active caspase-1 enzyme through autoactivation [102]. The active caspase-1 can then break down gasdermin D into its lytic N-terminal fragment to make holes in the plasma membrane to kill the cell. Indeed, gasdermin D knockout hearts were protected against acute cardiac IRI [103]. This process of cell death has been called pyroptosis, and because of its significance in acute cardiac IRI, it can be considered a high-potential target for reperfusion injury therapy.

## 4. Translating Preclinical Cardioprotection into the Clinical Arena: Role of Risk Factors, Comorbidities, Comedications, Peri-Operative Care, and Ischemia Duration

One of the main obstacles to successful translation from the preclinical towards the clinical condition has been the omission of patient risk factors, comorbidities, comedications, and peri-operative treatments in preclinical models of cardiac IRI. Risk factors and comorbidities (aging, diabetes, hyperglycemia, sex, metabolic syndrome, and hypertension), comedications (statins, β blockers, metformin, GLP-1 agonists, and SGLT2is), and peri-operative treatments (heparin, aspirin, P2Y12 platelet inhibitors, nitroglycerine, opioids, benzodiazepines, and propofol) can all abrogate cardioprotective interventions. Most of these factors have been recently summarized in comprehensive reviews [1,104]. Interestingly, recent work revealed a possible mechanism by which comorbidities such as prediabetes, hyperglycemia, and metabolic syndrome may abrogate cardioprotective interventions such as insulin treatment or ischemic preconditioning [105]. This mechanism entailed enhanced nitration of caveolin-3 at tyrosine 73 by peroxinitrite formed from increased NADPH oxidase-induced ROS and increased iNOS-induced NO, thereby disruption the Cav3 signalosome needed for insulin sensitivity, resistance to myocardial ischemia, and several cardioprotective signaling pathways [105]. Therefore, decreasing nitration through inhibition of iNOS or NOX activity, e.g., through the use of SGLT2 inhibitors that have been reported to reduce ROS and increase NO [106,107], may help to restore the cardiac intrinsic pathways of protection. 

An additional, often neglected factor that also determines the effectiveness of certain protective interventions, but largely deviates between preclinical and clinical cardiac IRI, relates to the duration and/or severity of the ischemic insult [108]. Whereas in preclinical models duration of ischemia is most often between 25 and 50 min with 100% obstructed coronary flow, in clinical models, the ischemic insult before the start of clinical treatment commonly lasts between 150 and 250 min of still-lingering (<10%) coronary flow [109,110,111,112]. Although for both infarcts, the size amounts to approximately 50% of the area at risk, it is likely that underlying cellular mechanisms causing cell death differ between short-term no-flow and long-term low-flow conditions. In addition, the sensitivity of cardioprotective interventions depending on ischemia duration may differ. For example, reducing ischemic-induced oxidative stress and boosting eNOS activity mostly provide protection against the short duration of no-flow ischemia (<30 min), whereas ischemic preconditioning, postconditioning, mPTP inhibition, opioids, sevoflurane, and metoprolol mainly protect against no-flow ischemia of 30–60 min duration. Conversely, mPTP inhibition and postconditioning can even increase IRI when applied to combat short no-flow ischemia, whereas none of these cardioprotective interventions protected against no-flow ischemia of >60 min [108]. From this perspective, devising a multitarget protective reperfusion strategy may be optimal [113]. Once an optimal strategy has been developed, step-by-step criteria for IMproving Preclinical Assessment of Cardioprotective Therapies (‘IMPACT’) should be met to improve the likelihood of translating novel cardioprotective interventions to the clinical setting [114]. 

## 5. Volatile Anesthetics and Noble Gases for Cardioprotection against IRI

### 5.1. Volatile Anesthetics

Nowadays, the volatile anesthetics isoflurane, sevoflurane, or desflurane are generally used for balanced anesthesia in the clinical setting. Cardioprotective properties of isoflurane were first described by Warltier et al., who showed infarct size reduction in the dog heart in vivo in the isoflurane-treated group [115]. Subsequently, several studies demonstrated cardioprotection by isoflurane against IRI by preconditioning and/or postconditioning [116,117,118]. Experimental studies tried to identify potential comorbidities, as well as concomitant use of medications in their models. Infarct size reduction by isoflurane was shown to be abolished by aging, but protection was restored by employing the antioxidant TEMPOL. Interestingly, when autophagy and mitophagy were inhibited, the cardioprotective effect of isoflurane was blocked not only in young rats but also in old rats treated with TEMPOL [119]. The importance of mitophagy and autophagy in the context of cardioprotective interventions in myocardial infarction is not yet clear [120]. The release of ROS generated after IR is inhibited by mitophagy. Furthermore, mitophagy plays a role in conditioning strategies but decreases with age, making mitophagy an interesting target for cardioprotective interventions in the aged myocardium [121]. Isoflurane-induced preconditioning inhibits cardiomyocyte autophagy by phosphorylation of p38 MAPK accompanied by a decreased expression of nucleotide-binding oligomerization domain containing 2 (NOD2) [122]. Autophagy protects against IRI; however, exaggerated autophagy has a decisive role in reperfusion injury-mediated myocardial dysfunction [120].

In the past, research on cardioprotection focused on mitochondria as a key element in the signal transduction pathway of conditioning strategies. Regarding conditioning effects on mitochondrial function, Xu and colleagues showed, irrespective of mitochondrial NO production, mitochondrial state 2 respiration uncoupling with diminished state 3 respiration by isoflurane [123]. As a part of the mPTP, Cyclophilin D (CypD) regulates mPTP by controlling the opening dimension of the pore [124]. In wild-type and CypD knockout mice, mitochondrial state 3 respiration was blocked by isoflurane, and ADP consumption was improved [125]. Isoflurane-induced cardioprotective properties were mediated by microRNA-21 through the signaling pathway containing Protein kinase B (Akt), NOS, and mPTP [126,127]. In contrast to microRNA-21 upregulation, microRNA-23 was shown to be inhibited by isoflurane [128]. Inhibition of microRNA-23 by isoflurane led to the protection of cardiomyocytes against oxidative stress [128]. The role of microRNAs in cardio- protection seems to be diverse and further research is needed to clarify the particular impact.

Cardioprotective effects induced by sevoflurane [129,130] are mediated by different pathways, i.e., Janus kinase (JAK) and signal transducers and activators of transcription (STAT) [131,132]. Janus kinase (JAK) and signal transducers and activators of transcription (STAT) are involved in sevoflurane-induced cardioprotection. JAK/STAT belong to the SAFE pathway. Sevoflurane-induced protection against reperfusion injury influences apoptosis via the SAFE pathway, and postconditioning by sevoflurane was abolished by a selective JAK2 inhibitor [133]. Besides effects on apoptosis, sevoflurane was also shown to have inhibitory effects on autophagy [134], mediated by reducing phosphatidylinositol 3-kinase catalytic subunit type 3 (PI3KC3). Sevoflurane-induced cardioprotection also restores autophagic flux impaired by IR, and this effect was NO-dependent [135]. The protective effects were completely abrogated in the presence of the NOS inhibitor L-NAME. The same results were observed following the administration of chloroquine—a blocker of autophagic flux [135]. Besides the RISK and/or SAFE pathway, other targets involved in cardioprotection were identified, i.e., vascular endothelial growth factor receptor (VEGFR), consisting of three subtypes, namely VEGFR-1, VEGFR-2, and VEGFR-3. VEGFR-1 was shown to be increased by sevoflurane [136], leading to myocardial preconditioning and decreased inflammation [136]. Furthermore, a specific inhibitor of VEGF-1, macrophage migration inhibitory factor-1 (MIF-1), abrogated the protective effect of sevoflurane [136], similar to isoflurane; in sevoflurane-induced protection microRNAs (miRNA) are also involved. A decrease in miRNA-155 expression induced by sevoflurane was associated with an increase in sirtuin 1 (SIRT1), thereby reducing infarct size and inhibiting cardiomyocyte apoptosis [137]. 

Notably, many patients that might need perioperative cardioprotection suffer from concomitant diseases, i.e., diabetes mellitus. Sevoflurane preconditioning-induced infarct size reduction was not affected in diabetic mice [138], an effect mediated by an AMP-activated protein kinase (AMPK)-independent activation of pro-survival mitogen-activated protein kinase (MAPK) members, whereas in non-diabetic mice, cardioprotection was AMPK-dependent [129]. Aging blocks sevoflurane-induced preconditioning [139] via activation of nuclear transcription factor kappa B (NFkB) regulated genes [139]. Promising cardioprotective effects of volatile anesthetics in diseased myocardium should be followed up by elucidation of the different and/or impaired underlying mechanisms in future research.

### 5.2. Noble Gases

Volatile anesthetics share some common pathways with other gases, e.g., noble gases. Within the group of six noble gases, xenon and helium have especially gained a great deal of interest regarding their cardioprotective properties [131,140,141,142,143,144]. Those so-called “inert” gases display profound biochemical activity and reduce infarct size in a rabbit IRI model [145]. Noble gases protect the myocardium when given before, during, or even after ischemia [131,140,143]. In contrast to xenon, helium is not an anesthetic, and thus, the cardioprotective effects are not linked to a hypnotic effect of the noble gases. 

Xenon has direct effects within the myocardium [131], including differential expression and phosphorylation of a variety of proteins, as well as blockade or activation of various channels. One major target of xenon-induced preconditioning of the heart is protein kinase C (PKC). The phosphorylation and subsequent translocation of the isoform PKC-Ɛ is a key mediator of xenon-induced cardioprotection [146]. Further downstream of PKC-Ɛ, the p38 MAPK and MAPK-activated protein kinase 2 (MAPKAPK-2) are regulated by xenon and connect its preconditioning properties to the cardiac cytoskeleton and actin stress fiber regulation via the small heat shock protein HSP27 [146,147]. Additionally, in vivo studies in rats using 3 × 5 min of xenon conditioning before IRI revealed that ERK (p44/42 MAPK) and the p54/46 MAPK (SAPK/JNK) are differentially regulated by xenon. Only ERK could be identified as a mediator in the cardioprotection by xenon [148]. The activation of PKC has been shown to be initiated by the regulation of mitochondrial KATP channels and PDK-1 in rats and rabbits in vivo [149]. 

All above-mentioned studies applied early cardioprotection protocols; however, xenon was also shown to be a potent inducer of late cardioprotection in vivo: xenon inhibited progressive adverse cardiac remodeling, contractile dysfunction and reduced the expression of β-myosin heavy chain and periostin proteins up to 28 days after a 60 min coronary artery occlusion in rats [150]. A total of 90 min of right ventricular ischemia followed by 120 min of reperfusion in a porcine model, whereby xenon was administered throughout ischemia and reperfusion, increased right ventricular afterload and myocardial contractility. On the cellular level, mRNA expression of type B natriuretic peptide (BNP) was hampered in the remote area of the left ventricle by xenon [151]. Another potential mediator of xenon-induced late preconditioning was shown in an in vivo late preconditioning model of rats using the COX-2 inhibitor NS-398: cyclooxygenase 2 (COX-2). Xenon, however, did not regulate mRNA expression of COX-2 in these animals, suggesting that xenon-induced late preconditioning is mediated most likely by an increased activity of existing COX-2 rather than a transcriptional upregulation of COX-2 mRNA [152]. 

Helium is already in clinical use for the treatment of respiratory diseases, and it can be easily applied to awake patients suffering from IRI. Inhalation of three times five minutes of 70% helium before IR reduced infarct size in rabbit hearts [145]. In follow-up studies, the authors administered several different inhibitors blocking a variety of signaling pathways described to be involved in volatile anesthetic and ischemic preconditioning of the heart. Phosphatidylinositol-3-kinase (PI3K), mitogen/extracellular signal-related kinase 1 (MEK-1), the 70-kDa ribosomal protein s6 kinase (p70s6kinase), ROS, KATP channels, and NO are mediators of helium-induced cardioprotection [145,153,154]. Inhibition of glycogen synthase kinase-3beta (GSK-3β) activity and activated apoptotic protein p53 degradation in a model with one, three, or five cycles of helium preconditioning reduced infarct size even more, suggesting a lowered threshold when these proteins are blocked. Most importantly, protection was completely reversed by mPTP opening with atractyloside during helium preconditioning [155]. Cyclosporine A, a selective inhibitor of mPTP, counteracts the blockade of helium conditioning in the presence of mild acidosis during the reperfusion in vivo [156]. Furthermore, morphine, in combination with helium, reduces infarct size after IR, whereby morphine alone had no effect on infarct size in an in vivo rabbit model [157]. As the effect could be inversed by blocking opioid receptors with naloxone, the involvement of these receptors has been suggested [157]. Even a low concentration of 30% helium in a singular dose 24 h before an ischemic insult was cardioprotective, and this effect was mediated most likely by COX-2 [158]. Regarding helium post-conditioning in healthy animals, 15 min of helium inhalation already protected the myocardium, but a prolonged inhalation for up to 60 min had no effect [159]. 

Most of these early studies on helium-induced cardioprotection did not investigate direct effects on the expression, phosphorylation, or activation of proteins. Oei et al. identified several important genes that where either up- or down-regulated by 15 min of helium postconditioning. The authors used gene expression arrays and found genes regulating necrosis, apoptosis (pro- and anti-), and autophagy either up- or down-regulated, showing a distinct pattern of gene regulation in the myocardium during helium postconditioning [160]. 

The central targets for helium-induced cardioprotection studied are mitochondria and the mPTP. Heinen et al. focused on mitochondrial calcium-sensitive potassium (mKCa) channels and changes in mitochondrial respiration by helium [161]. By measuring the rate of oxygen consumption in isolated mitochondria of young and old rats after helium conditioning, they showed that helium reduced the respiratory control index (state 3/state 4) only in young animals. Furthermore, the blockade of mKca channels by Iberotoxin abolished the protective effect of helium on infarct size reduction and respiratory control only in young animals [161]. 

In young and old rats, PKA was reported as an upstream target of mKCa channels. Interestingly, activation of mKCa channels in both young and old animals by NS1619 reduced infarct size, pointing to a pivotal role of these channels also in aged myocardium [162]. The adenylyl cyclase activator forskolin in a dosage of 300 μg/kg was only able to reduce infarct size in young animals. However, a much higher dosage of 1000 μg/kg was effective even in old rats, suggesting that the upstream regulation of mKCa channels by PKA might be a critical step for the age-dependent loss of helium-induced cardioprotection [162]. 

The influence of hypertension on helium conditioning was investigated in spontaneously hypertensive rats that were subjected to 25 min of ischemia followed by 120 min reperfusion using three different conditioning protocols in vivo. One group of animals received 70% helium post-conditioning (15 min after the index ischemia), one group received post-conditioning in combination with helium late preconditioning (application of helium 24 h before the IR procedure), and the third group received an additional 3 × 5 min cycle of preconditioning shortly before IR on top of the second group [163]. Only the triple intervention could effectively protect spontaneously hypertensive rats against IRI. 

Diabetic disorders prevent helium-induced early preconditioning and postconditioning [163]. In pre-diabetic obese Zucker rats, no helium-induced mitochondrial uncoupling could be observed. On the contrary, in the non-diabetic Zucker lean rats, mild mitochondrial uncoupling of oxygen consumption was detected [163]. Among the molecular targets investigated in this study, helium only affected GSK-3β phosphorylation in Zucker lean animals. Other important targets, such as ERK 1/2 and Akt phosphorylation, were surprisingly not regulated in both animal types [163]. 

Helium affects many intracellular proteins without receptor binding in the heart. Caveolins, small proteins that are embedded in the Caveolae of the cell membrane, are supposed to be a target for Helium. As Caveolins have a scaffolding domain that has been described to bind several proteins involved in helium-induced cardiac protection (e.g., Src kinases, PI3K, eNOS, PKC isoforms, and ERK), a regulation of Caveolins by helium seems likely [164,165,166]. 

Two of the isoforms of Caveolin (Caveolin 1 and 3) are critically involved in helium postconditioning [167]. After helium administration, an increased level of Caveolin 3 was found in the plasma of rats. Furthermore, both isoforms were upregulated in the infarcted area of rat hearts [167]. Short (5 min) administration of helium before cardiac arrest leads to differential expression patterns of both isoforms in the myocardium and reduced cardiac apoptosis [168]. 

Employing isolated Langendorff-perfused mice hearts, cardioprotection by helium could not be detected [169]. The lack of any circulating blood in the Langendorff system was assumed to be one explanation for this surprising outcome. In follow-up experiments, the membrane fractions showed a decrease in Caveolin 1 and 3 expressions, whereby both isoforms were elevated in the platelet-free plasma of the mice [169]. Secretion of caveolins into the blood stream following helium inhalation is suggested. 

In isolated neonatal rat cardiac fibroblasts, helium conditioning induces migration of cardiac fibroblasts and might thereby mediate cardioprotection. However, this effect was not mediated by an increased release of extracellular vesicles (EV). Helium decreased the amount of medium EV [170]. The cardioprotective pathways that have been discussed for helium in this section can be found in Figure 2.

To summarize, convincing evidence supports that the biological ‘inert’ gases xenon and helium affect several molecular pathways in the heart, which contribute to the observed cardioprotection against IRI of both gases in the experimental setting.

## 6. Translating Preclinical Cardioprotection by Volatile Anesthetics and Noble Gases into the Clinical Arena

Although experimental findings on volatile anesthetic-induced cardioprotection were very convincing, clinical evidence for these protective effects is lacking [171,172,173,174,175,176,177]. Anesthesia with volatile anesthetics was shown to reduce mortality in cardiac surgical patients but not in patients undergoing non-cardiac surgical interventions [178]. Likhvantsev et al. compared sevoflurane with propofol in nearly 900 patients undergoing cardiac surgery. Sevoflurane reduced one-year mortality and length of hospital stay [171]. However, in a similar patient population, Landoni et al. could not demonstrate any effect on mortality and/or hospital stay when comparing sevoflurane with propofol [172]. The same author confirmed his results later in a large multicenter trial including 5400 cardiac surgical patients showing no benefits on one-year mortality when using isoflurane, sevoflurane, or desflurane compared to propofol [172]. Furthermore, other outcomes, such as myocardial infarction, did not differ between the anesthetic regimens [173]. In non-cardiac surgery, patients with increased cardiovascular risk (*n* = 385), no protective effect of sevoflurane versus propofol on myocardial ischemia was observed [174]. Particularly with regard to patients undergoing cardiac surgery, where balanced anesthesia has been re-commended for years, current research shows no evidence in favor of volatile anesthetics compared to intravenous anesthetics. 

Similar results were reported for noble gas-induced cardioprotection in the clinical scenario.

In healthy volunteers, employing a forearm blood flow model to investigate endothelial function, helium reduced post-ischemic endothelial dysfunction without affecting plasma levels of cytokines, adhesion molecules, or microparticles known as mediators of helium-induced organ protection [179]. In patients undergoing coronary artery bypass surgery, neither helium pre-conditioning (3 × 5 min helium inhalation before aortic cross-clamping) nor post-conditioning (helium inhalation at the start of reperfusion) reduced postoperative troponin release [180]. Even a combination of pre- and postconditioning had no cardioprotective effect. 

Contrary to hypothermia alone, xenon, in addition to hypothermia, attenuated myocardial damage in patients after out-of-hospital cardiac arrest and ROSC [181]. In addition, xenon combined with hypothermia was associated with greater recovery of left ventricular systolic function in comparison with hypothermia alone, indicating some cardioprotective properties of xenon in this clinical setting [182]. 

A multicenter, international, and randomized clinical trial assessed the cardioprotective effects of xenon anesthesia in patients undergoing coronary artery bypass graft surgery as compared to sevoflurane- or propofol-based anesthesia [176]. In 492 patients receiving either propofol, sevoflurane, or xenon, a reduction in troponin I release was shown between the xenon and propofol and between the sevoflurane and propofol group. The difference in troponin release was small, and no conclusion can be drawn whether the observed effect is clinically relevant. 

To summarize, similar to volatile anesthetics, a clinically relevant cardioprotective effect of helium or xenon is absent.

It has to be mentioned, however, that also other medical gases have been investigated for their cardioprotective effects. Amongst these, NO and also Hydrogen (H2) gas have been found effective in protecting against IRI of the heart in different animal studies [183,184,185,186,187]. In a double-blind, randomized, placebo-controlled phase II trial, the impact of inhaling 80 ppm NO as an adjunctive therapy before percutaneous coronary intervention was investigated. The effects of NO were observed for up to 4 h following reperfusion. While the inhalation of NO did not show a significant reduction in infarction size (measured by magnetic resonance) between 48 and 72 h compared to the placebo group, a trend towards improved left ventricular (LV) functional recovery was observed with NO inhalation [188]. Furthermore, experimental studies have investigated the role of hydrogen gas in preventing ischemia/reperfusion injury of the heart and other organs [187] for a recent review in this area [189]. Ohsawa et al. discovered that hydrogen gas has a selective antioxidant effect that helps to scavenge harmful ROS, such as hydroxyl radicals (OH·), while preserving the beneficial signaling functions of other ROS [190]. These features help to reduce oxidative stress and limit damage to cardiac tissues. Additionally, hydrogen gas was proven to suppress the inflammatory responses by modulating pro-inflammatory cytokines and, therefore, can help prevent further damage to the heart during ischemia/reperfusion injury [189].

The pathways per drug group that have been shown to be involved in preventing IRI of the heart have been summarized in Table 1. 

## 7. The Endothelium as Target to Protect against IRI during Cardiac Surgery with Cardiopulmonary Bypass

One of the most common surgical interventions in which global cardiac ischemia is induced is cardiac surgery. During cardiac surgery, cardiac ischemia is induced by cross-clamping of the aorta, as most surgeons prefer a non-beating heart with a blood-free operating field. During cross-clamping of the aorta, coronary artery perfusion is ceased, and ischemia is induced. To maintain systemic blood circulation, blood is transferred to a heart-lung machine, e.g., by extracorporeal cardiopulmonary bypass (CPB). 

Since the start of cardiac surgery in the 1950s, considerable research has been performed to reduce the damaging effects of aorta cross-clamping during cardiac surgery. Cardioplegic strategies have been the subject of many recent reviews [191,192], but also anesthetic agents, as discussed above, have been shown to protect the heart against IRI. 

Myocardial IRI is mainly characterized as a disease of cardiomyocytes, but other cellular compartments, such as the vasculature, also play a role. Most of the non-cardiomyocyte cells consist of endothelial cells [193]. Endothelial cells constitute the inner lining of arteries, veins, and capillaries and form a barrier between vessels and tissue, regulate blood flow, and are essential for tissue delivery of oxygen and nutrients. Cardiomyocytes have received the most attention as they are more sensitive to IRI compared to endothelial cells; however, with increasing knowledge of cardiac cellular composition, it has become evident that crosstalk between cardiomyocytes and endothelial cells is crucial for cardiac function [194]. Besides myocardial IRI, also endothelial dysfunction is a well-known phenomenon in cardiac surgery patients. Patients undergoing cardiac surgery with CPB have disturbed capillary perfusion [195], resulting in hypoxia and ischemia. This is, among other causes, due to a systemic inflammatory response and activation of the endothelium, leading to increased permeability of the endothelium, fluid accumulation, and tissue edema [196], thereby hampering oxygen exchange. A healthy endothelium is important for cardiac function [197], and endothelial cells can improve cardiomyocyte survival after ischemia [198]. Moreover, pharmacological targeting of the endothelium might be an effective way to protect the heart during IRI, given its capacity to release protective factors [199]. 

### 7.1. Oxidative Stress

The systemic inflammatory response during CPB is associated with increased oxidative stress and ROS formation [200]. Used antioxidants during CPB are, among others, L-arginine and N-acetyl-cysteine (NAC). Supplementation of L-arginine, a precursor for NO synthesis, in cardioplegia solution, increased NO levels and attenuated ROS radical-mediated myocardial injury in patients undergoing CABG with CPB [201]. NAC administration reduced ROS formation and creatine kinase-MB in cardiac surgery patients on CPB [202], whereas, in another study in the same patient population, NAC decreased oxidative stress substantially; however, it did not improve cardiac troponin I level [203]. Although promising results, the cardioprotective effects of antioxidants via reducing oxidative stress need further investigation.

Administration of NO, NO donors, or drugs that enhance NO release prior to ischemia protects against myocardial IRI. Drugs that enhance NO release are statins, calcium antagonists, angiotensin-converting enzyme (ACE) inhibitors, and dexamethasone, which are not discussed as these drugs are already used in the clinical setting. Supplementation of the NO donor S-nitroso human serum albumin prevented eNOS uncoupling and improved myocardial perfusion and function in a pig CPB model [204]. However, this drug has not been studied in the clinical setting. Interestingly, the NO donor Nicorandil protected the heart via the reduction in oxidative stress in a rabbit CPB model [205] and reduced inflammation and troponin T levels in patients undergoing cardiac surgery with CPB [206,207]. In addition, nitroglycerin administration in cardioplegia solution increased NO levels and decreased troponin T levels in CABG patients [208]. In contrast, treatment with glyceryl trinitrate did not affect cardiac troponin T levels and even abrogated the cardioprotective effect of RIPC in cardiac surgery patients [209]. Taken together, NO donors seem promising in protecting the heart via the vasculature during CPB; however, it should be taken into account that most NO donors have platelet inhibitory properties and might therefore have clinical consequences in terms of postoperative bleeding [210]. 

### 7.2. Glycocalyx

The surface of the vascular endothelium is covered by a gel-like layer called the endothelial glycocalyx, which regulates vascular resistance, permeability, and leukocyte recruitment. IR severely damages the endothelial glycocalyx, which can be detected by increased circulating levels of its principal constituents, syndecan-1 and heparan sulfate [211,212]. Degradation of the glycocalyx can result in myocardial edema and microvascular obstruction in patients following myocardial infarction [213]. Glycocalyx damage is also reported following aortic cross-clamping in rats [214] and in cardiac surgery patients [215,216]. During remote ischemic conditioning, glycocalyx thickness was improved in patients after myocardial infarction [217], and anesthetic conditioning protects the glycocalyx layer. In isolated pig hearts, sevoflurane protected the post-ischemic heart against endothelial dysfunction by reducing glycocalyx degradation [218]. Sevoflurane decreased glycocalyx degradation in patients undergoing heart valve surgery with CPB compared to propofol anesthesia [219]. Thus, pharmacological conditioning might preserve the endothelial glycocalyx. 

Research on the effects of pharmacological conditioning on the glycocalyx during cardiac surgery is sparse. Lidoflazine, a calcium-channel blocker, preserves the glycocalyx and protects cardiac function in patients undergoing multiple aorta-coronary bypass grafting [220]. Although it has been known for many years that reperfusion results in cardiomyocyte calcium overload, as described above [221], calcium-channel blockers have not been followed up in the clinical setting. However, from a vasculature perspective, combination therapy with a calcium channel blocker could be of interest.

More recently, matrix metalloproteinases have been shown to be commonly upregulated in cardiac surgery patients with CPB [222] and are suggested as a potential mechanism triggering glycocalyx damage. Prophylactic treatment with doxycycline reduced glycocalyx damage in cardiac surgery patients with CPB by inhibiting matrix metalloproteinases [223]; however, it did not ameliorate cardiac mechanical dysfunction following reperfusion [224]. 

Other pharmacological means to reduce glycocalyx degradation during cardiac surgery have been explored in the context of CPB coating and priming fluids. Interestingly, the glycocalyx layer was better preserved when the CPB circuit was coated with heparin compared to phosphorylcholine in cardiac surgery patients [225]. In contrast, glycocalyx degradation did not differ between priming the CPB circuit with human albumin or gelofusine in rats [226]. Unfortunately, the above-mentioned studies did not investigate whether the protection of the endothelial glycocalyx reduced myocardial IRI. Interestingly, treatment with recombinant syndecan-1 reduced glycocalyx damage and partly ameliorated cardiomyocyte damage in mice with myocardial IRI [227], linking endothelial glycocalyx and cardiomyocytes during myocardial IRI.

### 7.3. Endothelial Barrier Function

Endothelial cells form a unique semi-permeable barrier for the transfer of solutes. Endothelial barrier function is regulated by cell–cell and cell–matrix adhesion, as well as endogenous mediators. Endothelial cells are joined together by adherens junctions, tight junctions, and gap junctions, and are anchored to the basement membranes via transmembrane receptors. Several types of endothelial receptors have been associated with the regulation of endothelial barrier function, such as VEGFR, sphingosine-1-phosphate (S1P) receptor, tyrosine-protein kinase (Tie2) receptor, and protease-activated receptor (PAR), and targeting them may open new therapeutic approaches to protect against myocardial IRI [228]. 

The involvement of VEGFR in cardioprotection was already suggested in the experimental setting during ischemic [229,230] and anesthetic conditioning [136,231]. The relationship between VEGF-A and the heart is interdependent as VEGF-A can activate cardiomyocytes, but upon inflammation, cardiomyocytes produce VEGF-A [232]. In a CPB rat model, VEGF protein is increased, especially following cardioplegia reperfusion [233]. In contrast, circulating VEGF-A levels decreased after CPB in cardiac surgery patients [234]. Interestingly, infusion of VEGF was protective for kidneys in beagles in CPB through the improvement in renal perfusion [235], but no studies have been performed yet on the effect on the heart during cardiac surgery. 

Additionally, S1P signaling is well known in IRI [236]. Targeting this signaling pathway not only protects the heart but also the vasculature. In isolated rat hearts, S1P decreased infarct size and myocardial edema [237]. Reduction in myocardial water content could not be explained by protection of the endothelial glycocalyx nor hemodynamic changes, but by the involvement of S1P in endothelial permeability. However, S1P receptor subtypes have contrasting effects. Subtype 1 (S1P1) is required in the protection of endothelial barrier function, whereas the activation of S1P2 and S1P3 receptors disrupts the endothelial barrier. Pharmacological activation of the S1P1 receptor with FTY-720 and SEW2871 preserved vascular function in rats on CPB [238] and might be an interesting target to protect both the vasculature and the heart as low circulating levels of S1P were found in cardiac surgery patients [239]. 

The endothelial angiopoietin/Tie2 system is one of the most important mechanisms in the regulation of endothelial barrier function and is dysregulated in cardiac surgery patients [240,241]. In short, angiopoietin-1 binds to the Tie2 receptor, thereby maintaining endothelial barrier function, whereas angiopoietin-2 binds antagonistically to Tie2, increasing endothelial permeability. Targeting Tie2 has been proposed as a promising strategy to improve vascular function, due to its key position in the regulation of endothelial barrier function. Indeed, pharmacological activation of Tie2 by vasculotide protected the vasculature in a CPB rat model [242]. The Tie2 antagonist angiopoietin-2 was highly expressed in endothelial cells at the infarct border zone after myocardial infarction in mice [243]. Moreover, endothelial-derived angiopoietin-2 was involved in vascular leakage and glycocalyx degradation, thereby worsening myocardial hypoxia [243]. Inhibition of angiopoietin-2 substantially ameliorated postischemic cardiovascular remodeling [243]. Furthermore, inhibition of angiopoietin-2 in cultured cardiomyocytes enhanced the cardioprotective effects of fibroblast growth factor 2 [244]. Taken together, these data suggest that combination therapy of angiopoietin-2 inhibition together with Tie2 activation has the potential to reduce myocardial IRI.

Another pharmacological agent with protective effects on endothelial barrier function is imatinib mesylate [245]. Imatinib is a tyrosine kinase inhibitor developed to treat Bcr/Abl-expressing leukemias, but Abl kinases are also involved in endothelial barrier regulation [246]. In a CPB model, imatinib prevented endothelial barrier dysfunction and attenuated pulmonary and renal injury [247]. Unfortunately, the heart was not studied, but in a rat model of acute myocardial infarction, imatinib reduced microvascular injury and myocardial infarct size [248]. As cardiotoxic effects are reported following treatment with another tyrosine kinase inhibitor [249], future research on the effect of several generations of tyrosine kinase inhibitors on IRI is required.

During cardiac surgery, CPB is associated with the generation of thrombin. Thrombin has adverse effects on the endothelium and on cardiomyocytes, independent of its procoagulant effects, and has therefore emerged as a possible mediator of IRI [250]. Thrombin can induce endothelial hyperpermeability via activation of the PAR1 thrombin receptor. During cardiac surgery, patients receive antifibrinolytics such as aprotinin or tranexamic acid. Aprotinin can, amongst others, prevent activation of the PAR1 receptor. In a rat CPB model, aprotinin preserved endothelial integrity and reduced edema in the kidney, but the effects on the heart were not studied [251]. In a CPB pig model, aprotinin preserved the loss of coronary adherens junctions, which resulted in the preservation of the endothelial barrier and reduced myocardial edema [252]. This was confirmed in cardiac surgery patients, were aprotinin showed cardioprotective effects [253,254]. However, contradictory results on the heart have also been reported [255]. Nowadays, the most commonly used antifibrinolytic is tranexamic acid. In a recent review and meta-analysis, the authors concluded tranexamic acid administration was associated with less myocardial injury in cardiac surgery patients [256]. These results suggest that direct thrombin inhibitors might be novel pharmacological agents to protect the heart. Indeed, one study showed cardioprotective properties during myocardial IRI [257], but future studies are needed to investigate these agents during cardiac surgery. A summary of the possible new therapeutics targeting the vasculature and thereby protecting the cardiomyocyte is given in Table 1 and Figure 3.

Although the role of the vascular endothelium as a dynamic regulator of tissue responses is increasingly recognized, the relation between vascular conditioning in the ischemic heart is underappreciated. The cross-talk between endothelial cells and cardiomyocytes provides an access point for therapeutic targeting and may be a promising approach to protect the heart against IRI.

The pathways per drug group that have been shown to be involved in preventing IRI of the endothelium are summarized in Table 1. 

## 8. Conclusions

In conclusion, while there are promising pharmacological approaches demonstrated in experimental studies for protecting the heart against IRI, there are challenges in translating these findings into the clinical setting. Several potential confounders have been identified within the last decennia, such as co-morbidities, co-medication, peri-operative care, and ischemic duration, and future studies should focus on understanding the underlying pathways involved in cardiac ischemia-reperfusion damage in order to evaluate the effectiveness of combination therapies targeting different cell types, such as endothelial cells and damage pathways. Overall, continued research in this area is critical for improving the outcomes of patients at risk of cardiac IRI.

## Figures and Tables

**Figure 1 cells-12-01432-f001:**
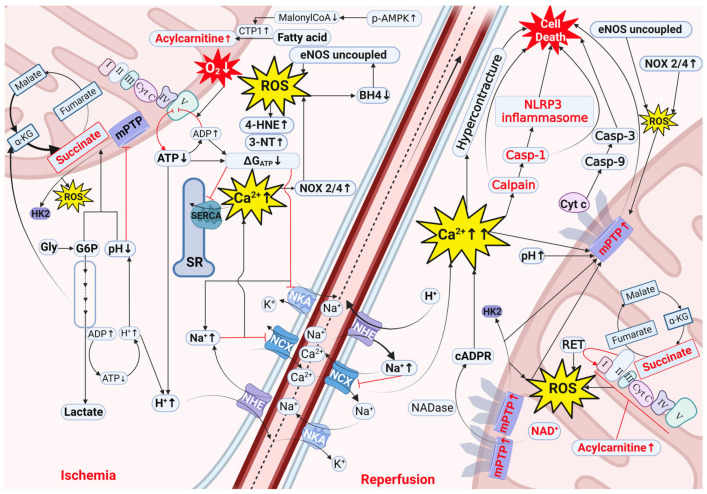
Schematic diagram of mechanisms during the ischemic and reperfusion period that together cause acute cardiac IRI.

**Figure 2 cells-12-01432-f002:**
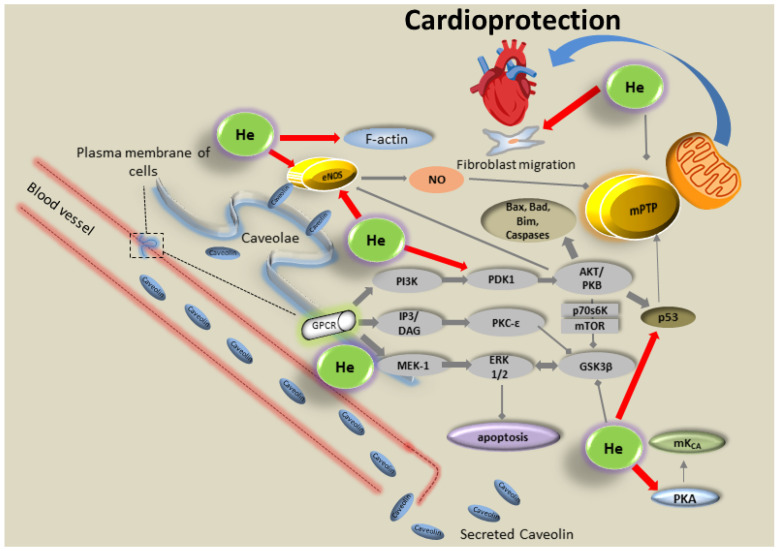
Schematic overview of Helium-induced cardioprotection: this diagram presents a summary of the established mechanisms involved in helium-induced cardioprotection, primarily through the RISK pathway, which is closely associated with changes in Caveolin-related processes. Helium is represented by a purple circle (He). Red arrows indicate activation or up-regulation, while squares denote suppression or down-regulation. Intracellularly, these mechanisms converge on the mitochondria, inhibiting the opening of the mitochondrial permeability transition pore (mPTP). Additionally, the potential pathway of “remote conditioning” by helium is illustrated on the left side of the diagram. Unknown and identified factors (potentially Caveolin, transported via exosomes) mediate protection in distant organs and enhance mitochondrial respiration in remote cells. GPCR = G-protein coupled receptor; MEK-1 = mitogen-activated protein kinase-extracellular signal-regulated kinase-1; ERK1/2 = Extracellular signal-regulated kinase 1/2; IP3 = inositol triphosphate-3; DAG = diacylglycerol; PKC-ε = protein kinase C epsilon; GSK3β = glycogen synthase kinase-3beta; PI3K = Phosphatidylinositol-3-kinase; PDK-1 = phosphoinositide-dependent protein kinase-1; PKB = protein kinase B; mTOR = mammalian target of rapamycin; P53 = Tumor protein P53; mPTP mitochondrial permeability transition pore; eNOS = endothelial nitric oxide synthase; NO = nitric oxide; L-NAME = L-NG-nitroarginine methyl ester; PKA = protein kinase A; mKCa = mitochondrial calcium-sensitive potassium channel; ROS = reactive oxygen species; Pi = inorganic phosphate; ATP = adenosine triphosphate; and ADP = adenosine diphosphate.

**Figure 3 cells-12-01432-f003:**
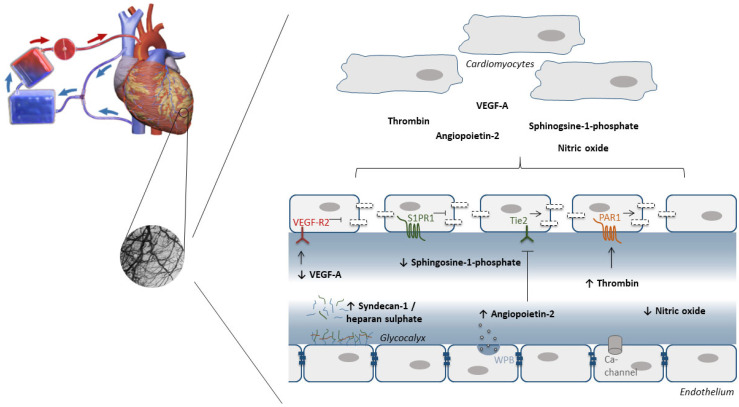
Overview of possible new therapeutics targeting the endothelium and thereby protecting the cardiomyocyte. Crosstalk between cardiomyocytes and endothelial cells is crucial for cardiac function. Endothelial cells constitute the inner lining of arteries, veins, and capillaries and form a barrier between vessels and heart. During cardiopulmonary bypass, the endothelium can be activated, leading to increased permeability of the endothelium (see top part of the schematic vessel), fluid accumulation, and tissue edema, thereby hampering oxygen exchange. Within the circulation, several proteins are up- or down-regulated during cardiopulmonary bypass. These circulating proteins can interfere with their receptors on the endothelium and also (in)directly affect the cardiomyocyte. They are, therefore, targets of interest to therapeutically protect the heart against IRI. VEGF-A = vascular endothelial growth factor A; VEGF-R2 = VEGF-receptor 2; S1PR1 = sphingosine-1-phosphate receptor 1; Tie2 = tyrosine kinase receptor; PAR1 = protease-activated receptor 1; WPB= Weibel–Palade bodies; Ca = calcium.

**Table 1 cells-12-01432-t001:** Promising drugs/compounds to reduce cardiac IRI and endothelium damage.

Drugs/Compounds	Mechanism	Setting	= > Phase II Trial?
Heart			
**NAD+ precursors** **(NR, NMN)**	restoring NAD levels	preclinical models of acute cardiac IRI	No
**Malonate**	reducing ROS at early reperfusion	preclinical models of acute cardiac IRI	No
**NLRP3 inflammasome inhibitors**	preventing inflammasome complex formation	preclinical models of cardiac IRI	No
**Caspase-1,4 inhibitors (VRT, emricasan)**	pyroptosis inhibition	preclinical models of acute cardiac IRI	No
**Calpain inhibitors**	proteolysis and pyroptosis inhibition	preclinical models of cardiac IRI	No
**Volatile anesthetics/** **Helium/Xenon**	activation of Survival pathways (RISK/SAFE), Caveolin 1/3, mitophagy, and autophagy	preclinical models of cardiac IRI	Yes
**Volatile anesthetics**	reduced microRNAs (miRNA-155), induction of VEGFR1	preclinical models of cardiac IRI	No
**Endothelium**			
**Sevoflurane**	activation of survival pathways (RISK/SAFE)	preclinical CPB models cardiac surgery patients with CPB	Yes
**Nitric oxide donors**	reducing oxidative stress	preclinical CPB models cardiac surgery patients with CPB	Yes
**Lidoflazine**	calcium channel blocker	patients undergoing multiple aorta-coronary bypass grafting	Yes
**Doxycycline**	inhibition matrix metalloproteinases	cardiac surgery patients with CPB	Yes
**(Indirect) Endothelial receptor activators**	strengthening endothelial barrier	preclinical CPB models cardiac surgery patients with CPB	Yes, but not all

NAD = nicotinamide adenine dinucleotide; NR = nicotinamide riboside; NMN = nicotinamide mononucleotide; IRI = ischemia-reperfusion injury; NLRP3 = NOD-like receptor family pyrin domain containing 3; ROS = reactive oxygen species; VRT = active component of the caspase-1/4 inhibitor VX-765; RISK = reperfusion injury salvage kinase; SAFE = survivor activating factor enhancement; VEGFR1 = vascular endothelial growth factor-receptor 1; and CPB = cardiopulmonary bypass.

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
