# Peer review of "Pharmacological Cardioprotection against Ischemia Reperfusion Injury—The Search for a Clinical Effective Therapy"

_cells, 2023, doi:10.3390/cells12101432_

Round 1
Reviewer 1 Report
The review is very well written about myocardial ischemia-reperfusion and potential clinical translation. There are new topics about the gas and endothelial protections. Some concerns are raised.
1) Generally, font in figures are too small.
2) Polyol pathway is a potential target to modulate sorbitol and NAD and aldose reductase inhibitor might be applicable for drugs.
3) NO/ROS balance is very important. In endothelium section, activation of eNOS by AKT and AMPK should be clarified and the potential drugs to modulate them. ENOS uncoupling is well written. Mitochondrial NOS or IK ATP could be also potential targets.Sodium nitrite might be discussed.
4) Gas section is very interesting. The potential of NO gas and hydrogen gas might be shortly discussed.
5) Thinking about the clinical situation, post-conditioning might be more applicable. Discuss about the post-conditioning, for example usage of lactate might be discussed.
Author Response
We attached a pdf letter in response to the comments. Please see attachement.

Reviewer 2 Report
I have read with great interest the reiview entitled "Pharmacological cardioprotection against ischemia reperfusion injury - the search for a clinical effective therapy". Overall, very interesting and meaningful. But several tips may help improve the manuscript:
1. In the begining of the review, it will be necessary if authors could provide an overview of other recent advances in the mechanisms of I/R injury, in addition to compounds included in this review.
2. It will be of great help if authors provide a table summarizing the names, mechanisms, limitations and other valuable terms of therapy targets discussed in the manuscript.
The English quality of this review is easy to understand.
Author Response
We attached a response letter to the comments of the reviewers. See attachement please.

Author Response
We attached a response letter to the comments of the reviewers. Please see attachement.
